# A reanalysis of the foundations of the macromolecular rate theory and an alternative based on chemical kinetics theory

Jinyun Tang[1], William J. Riley[1]

[1]Department of climate sciences, earth and environmental sciences area, Lawrence Berkeley National Laboratory, Berkeley, CA, USA.

*Correspondence to*: Jinyun Tang (jinyuntang@lbl.gov)

**Abstract.** The macromolecular rate theory (MMRT) has been proposed as a mechanistic scheme to describe the temperature dependence of enzymatic reactions, and has enjoyed quite some popularity recently. MMRT was motivated by assuming that enzyme denaturation is not sufficient to explain the decline of enzyme activity above an optimal temperature, and was derived with two experimental assumptions: (1) the half saturation parameter is independent of temperature; and (2) when the substrate concentration is kept at 10 times of the half saturation parameter at reference temperature, the enzyme assays are substrate saturated under all experimental temperatures. We show that thermally reversible enzyme denaturation could be essential to consistently interpret the temperature dependence of enzymatic reactions, and due to the temperature-dependence of the half saturation parameter, neither of the experimental assumptions of MMRT held. Consequently, the MMRT estimated temperature sensitivity of the maximum catalysis rate is inaccurate. It can mischaracterize temperature-related biochemical behaviors, such as inferring the existence of a unique optimal temperature where biochemical rate peaks, and the shift of this optimal temperature as an indicator of thermal acclimation or adaptation. We proposed a chemical kinetics theory that explicitly incorporates the observed thermally reversible enzyme denaturation, von Smoluchowski's diffusion-limited chemical reaction theory, and Eyring's transition state theory to interpret the temperature dependence of enzymatic reactions. Since the chemical kinetics theory performed equally successful in fitting the enzyme assay data used in deriving MMRT, and has incorporated more relevant empirical observations and well-established theories than MMRT, we recommend it as a better candidate for mechanistic modeling of the temperature dependence of biogeochemical rates. However, MMRT is still a better model than the conventional $Q_{10}$ and Arrhenius functions for describing the emergent temperature dependence of biochemical rates.

## 1 Introduction

Recently, the macromolecular rate theory (MMRT) has been proposed to interpret observations of enzyme-catalyzed chemical reactions (Hobbs et al., 2013). These rates often show a pattern that first increases gradually, then plateaus, and finally decreases rapidly with temperature. The authors of MMRT were motivated by asserting that "*denaturation is*

*insufficient to explain the decline in enzymatic rates above $T_{opt}$*", and proposed that the change in heat capacity associated

with enzyme catalysis and its consequent effect on the temperature dependence of the Gibbs free energy of activation can

describe the temperature dependence of enzyme activity. Following the success of Hobbs et al. (2013) on modeling the

temperature dependence of single-enzyme catalyzed reactions, Schipper et al. (2014) showed that MMRT is able to better than

the Arrhenius-like functions for fitting measured relationships between soil biogeochemical rates and temperature, including

those for aerobic respiration, methane oxidation, nitrification, and denitrification. Later, Alster et al. (2016) demonstrated that

MMRT was successful at capturing the temperature dependence of extracellular enzyme activities, including those of $\beta$-

glucosidase, leucine aminopeptidase, and phosphatase. Following these studies, Liang et al. (2018) recommended that MMRT

should be used for improved description of the measured relationship between plant leaf respiration and temperature. Recently,

Alster et al. (2020) advocated that MMRT should be used widely to represent the temperature dependence of many types of

soil biogeochemical processes. In spite of their success, these studies have not explained clearly what they meant by enzyme

denaturation and how MMRT can be logically extended from single-enzyme reactions to populations of biological cells that

carry out their metabolism using many enzymes.

The popularity of MMRT is built upon two observations: (1) MMRT is able to match measured relationships of

biochemical rates *versus* temperatures better than the popular Arrhenius-like functions and the $Q_{10}$ function, and (2) MMRT

parameters have more mechanistic meaning by involving thermodynamic definitions than other empirical functions that are of

similarly good descriptive power but with mechanistically less interpretable parameters (e.g., the log-polynomial function

(e.g., O'Sullivan et al., 2017), the four-parameter square root function (Ratkowsky et al., 1983), and see (Grimaud et al., 2017)

and (Noll et al., 2020) for more examples). Despite these merits, we show here limitations arising from the two experimental

assumptions used in their enzyme assays for developing MMRT (Hobbs et al., 2013): (1) the half saturation parameter is

independent of temperature (as implied by their assumption that the ratio of substrate concentration to half saturation parameter

was kept at two while using the same substrate concentrations under all temperatures for enzyme barnase and its mutant), and

(2) the constant value 10 for the ratio of substrate concentration to half saturation parameter at reference temperature ensures

that their enzyme assay system is substrate saturated under all experimental temperatures. (We note that Hobbs et al. (2013)

adopted their second experimental assumption from the enzyme assay protocol in Peterson et al. (2004), who proposed a four-

parameter (plus time) thermodynamically based equilibrium model that is able to fit the non-monotonic temperature dependent

enzyme reactions quite well). Instead, we find that by incorporating the well observed thermally reversible denaturation of

enzymes (i.e. the dynamic transition between their native folded state and the unfolded state as a function of temperature and

solution conditions (e.g., Oliveberg et al., 1995;Anfinsen, 1973)) into the chemical kinetics, we can satisfactorily explain the

non-monotonic temperature response of enzyme catalysis rate, while maintaining the logical consistency between theory and

the supporting empirical data.

Besides MMRT, a few other models with mechanistically interpretable parameters are also capable of equally well

interpreting the non-monotonic temperature dependence of enzyme modulated reactions, including growth rates. Notably,

Sharpe and Demichele (1977) proposed a model that incorporates the empirical observation of thermally reversible enzyme

denaturation and the transition state theory (Eyring, 1935). Specifically, they considered that enzymes are in reversible

transition between three states, one cold-induced inactive state, one heat-induced inactive state, and one active state which is

able to carry out the catalysis. By assuming reactions to be substrate unlimited, they obtained a model with five thermodynamic

parameters that is able to almost perfectly fit published temperature dependent growth rates of eight poikilothermic organisms

(see their Figures 5 and 6). (The applicability of the Sharpe-Demichele model to growth rates of an organism is based on the

assumed existence of control or master enzymes (Johnson and Lewin, 1946).) Motivated by the success of Sharpe and

Demichele (1977) and the work on thermally reversible protein denaturation by Murphy et al. (1990), Ratkowsky et al. (2005)

grouped the two inactive states into one, and, again assuming no-substrate limitation, derived a model with two thermodynamic

parameters and two enzyme informatic parameters, which was able to very accurately fit 35 sets of observed temperature

dependent bacterial growth rates. The model by Ratkowsky et al. (2005) was later used by Corkrey et al. (2012) and Corkrey

et al. (2014) to successfully interpret the temperature dependent growth rates of many more poikilothermic organisms. Ghosh

et al. (2016) extended the model by Ratkowsky et al. (2005) to include the thermally reversible denaturation of many enzymes

and proteins informed by proteomics, and were able to satisfactorily interpret the measured temperature-dependent growth

rates of mesophiles and thermophiles.

          The thermally-reversible enzyme denaturation occurs due to the thermal motion of molecules and ions in the solution

of enzyme proteins (Finkelstein and Ptitsyn, 2016). As thermal motion is ceaseless, according to Boltzmann's law in statistical

mechanics (Feynman et al., 2011), enzyme molecules will be distributed among different configurations that can be quantified by their respective energy status. Therefore, under any biologically feasible temperature, some of the enzyme molecules will not be in their biologically active native states. That is, at any life amenable temperature, only a fraction of enzymes is able to catalyze the corresponding biochemical reaction. Consequently, by not explicitly taking into account the thermally-reversible enzyme denaturation (or by assuming all enzyme denaturation are irreversible), we believe MMRT may have missed some important mechanistic insights on the temperature control of enzymatic reactions.

In the following, we first present our analysis of the assumptions involved in the development of MMRT. Then we describe an alternative interpretation of the observed non-monotonic relationship between biochemical rates and temperatures that is more consistent with protein physics and the theory of chemical kinetics. Finally, we discuss how our alternative formulation will lead to mechanistically more accurate representations of the temperature dependence of biogeochemical reaction rates.

## 2. Methods

### 2.1 The enzymatic reaction problem

The simplest form of enzymatic reactions as, described by MMRT (introduced in section 2.2) and the alternative theory (presented in section 2.3), can be formulated as:

$$E_n + S \underset{k_1^-}{\overset{k_1^+}{\rightleftarrows}} E_n S \xrightarrow{v_{max}} E_n + P, \qquad (1)$$

where $E_n$ is the concentration of free enzymes whose conformation structure is in the native state (i.e., being active and able to carry out the catalysis; here and below we will use "native" and "active" interchangeably according to the appropriateness of the context), $P$ is the concentration of product molecules, $S$ is concentration of the substrate, $E_n S$ is concentration of the enzyme-substrate complex, and $k_1^+$, $k_1^-$, and $v_{max}$ are temperature ($T$) dependent kinetics parameters. Although it is not necessary for the validity of the Michaelis-Menten kinetics (Briggs and Haldane, 1925), for scaling purpose, $v_{max}$ (the maximum enzymatic catalysis rate) is often assumed to be much greater than $k_1^-$ (Tang and Riley, 2017;Kooijman, 2009;Holling, 1959;Aksnes and Egge, 1991;Van Slyke and Cullen, 1914). Moreover, throughout this study, we take all variables to be in ISO units.

By applying the law of mass action and the quasi-steady-state-approximation to equation (1), we obtain the Michaelis-Menten equation for the overall reaction rate $F$:

$$F = v_{max} \frac{E_{nt}S}{K+S},$$

(2)

where $K = v_{max}/k_1^+$ is the half saturation parameter, and $E_{nt}$ is the total concentration of enzymes that are able to form enzyme-substrate complexes.

We next describe how MMRT and the chemical kinetics theory represent the temperature dependence of $F$.

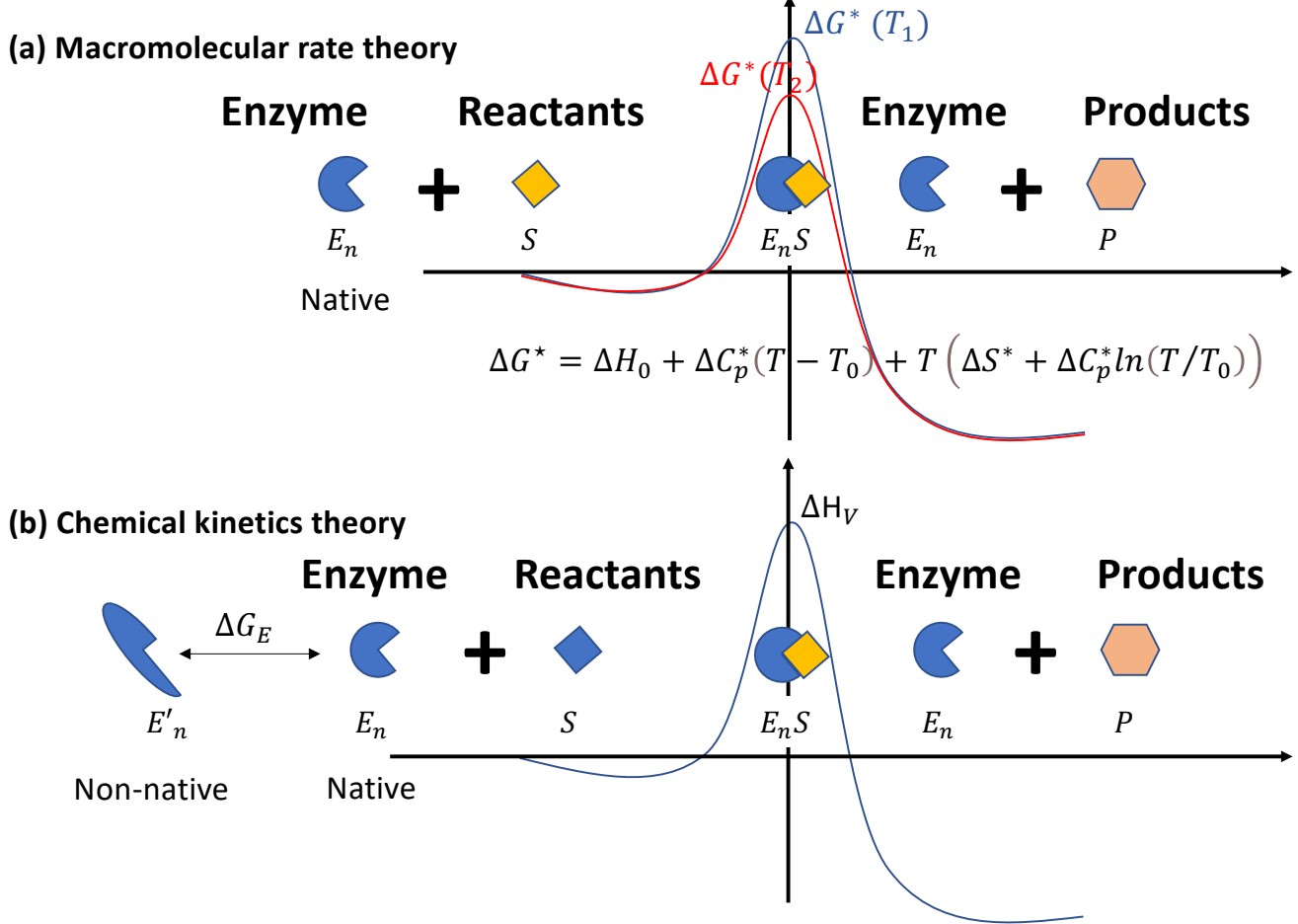

**Figure 1: (a)** In the macromolecular rate theory (MMRT), the Gibbs free energy of activation $\Delta G^*$ is a nonlinear function of temperature, giving rise to the non-monotonic temperature response of catalysis rate; **(b)** in the chemical kinetics theory, the Gibbs free energy of activation is a linear function of temperature, the enthalpy of activation $\Delta H_V$ is constant, and the thermally reversible denaturation of the enzymes leads to the non-monotonic temperature response of catalysis rate. Other variables are defined in the main text.

## 2.2 The macromolecular rate theory (MMRT)

MMRT (Figure 1a) applies the transition state theory (Eyring, 1935) to describe the maximum reaction rate $v_{max}$ (called $k(T)$ in the MMRT representation) as

$$k(T) = \frac{k_B T}{h} exp\left(-\frac{\Delta G^\star}{RT}\right), \tag{3}$$

where $k_B$ is the Boltzmann constant, $h$ is the Planck constant, $R$ is the universal gas constant, and $\Delta G^\star$ is the Gibbs free energy of activation (determined as the difference in Gibbs free energy between the ground state and the transition state). Most importantly, motivated by (La Mer, 1933) (as pointed out by an anonymous expert on MMRT who reviewed a previous version of this manuscript), MMRT assumes $\Delta G^\star$ to be dependent on temperature nonlinearly, such that

$$\Delta G^\star = \Delta H_0 + \Delta C_p^*(T - T_0) + T\left(\Delta S^* + \Delta C_p^* ln(T/T_0)\right), \tag{4}$$

where $\Delta H_0$ and $\Delta S^*$ are enthalpy and entropy between the ground and transition states at reference temperature $T_0$, respectively, and $\Delta C_p^*$ is (the change in) heat capacity associated with enzyme catalysis. Without invoking the enzyme denaturation process, MMRT effectively assumes that all enzymes are in their native state and are capable of forming complexes with the substrate molecules, and all enzyme-substrate complexes are active for generating products. It is through a negative $\Delta C_p^*$ that the temperature dependence of $k(T)$ becomes non-monotonic, in which it first increases from low temperature and then, after passing a peak reaction rate, falls off at high temperature. (However, recently, based on molecular dynamics simulations, Aqvist et al. (2020) and Aqvist and Van der Ent (2022) suggested that not all enzyme-substrate complexes are active and inferred $\Delta C_p^*$ to be zero. Moreover, for thermally reversible protein denaturation, Oliveberg et al. (1995) observed a negative heat capacity during their refolding into native states, which coincidently has the same sign as $\Delta C_p^*$ in MMRT. We will discuss this in detail later in section 4.)

Although Hobbs et al. (2013) did not explicitly state that the affinity parameter $K$ (aka $K_M$) in their fitting of MM kinetics is temperature independent, their analysis of enzyme assay data effectively assumed so by attributing all temperature dependence of $F$ to $k(T)$ and assuming all enzyme assays are substrate saturated. That is, MMRT computes $F$ as

$$F = F_{MMRT} = k(T)\frac{E_t S}{K_0 + S}, \tag{5}$$

where $K_0$ is $K$ empirically determined at the reference temperature $T_0$, and $E_t$ is the total concentration of the enzyme.

In the reanalysis using chemical kinetics theory below, we show that the temperature dependence of $K$ is related to

that of $v_{max}$, so that the MMRT-derived temperature dependence of $k(T)$ is a function of substrate concentration. In contrast, in the enzyme assays by Hobbs et al. (2013), for five out of seven enzymes, it was assumed that a substrate concentration of $\sim 10K_0$ at the reference temperature is sufficient to ensure each system is substrate saturated (i.e. $S/K \gg 1$) under all temperatures, so that the inferred $k(T)$ is independent of substrate concentration. Although Hobbs et al. (2013) never stated that $k(T)$ fully captures the temperature dependence of $F$, later applications (e.g., Schipper et al., 2014;Liang et al., 2018;Alster

et al., 2016) recommended MMRT to replace the popular Q10 function or Arrhenius function to represent the temperature sensitivity of biochemical rates in soils and plants. (We highlight that in those applications, what MMRT, the Q10 and Arrhenius functions represent are emergent temperature response dependent on the substrate and plant or soil conditions used in deriving the empirical data.) Alster et al. (2020) recognized the possible temperature dependence of $K$, but still considered the temperature dependence of $v_{max}$ to be captured in $k(T)$. Below we will show that $k(T)$ as determined by MMRT convolves

the temperature dependence of $v_{max}$ and $K$, so that it fails to capture the temperature dependence of $F$.

*2.3 The chemical kinetics theory*

Chemical kinetics theory (Figure 1b) incorporates the observation that a fraction $(1 - f_E(T))$ of the enzymes $(E_t)$ are in the thermally reversible denatured non-native state (e.g., Finkelstein and Ptitsyn, 2016;Ghosh and Dill, 2009), so that the total catalytically active enzymes is $f_E(T)E_t$. (As we argued in the introduction, such reversible transition between native and

unfold states is ensured by the ceaseless thermal motion of molecules and ions in the enzyme solution.) Further, by thermodynamics, Jin and Bethke (2003) showed that, besides the catalysis by enzymes, the chemical reaction is also driven by a thermodynamic-potential $\Delta G_R$ whose effect can be parameterized through a function $f_R(T)$, where $\Delta G_R$ is the Gibbs free energy of the chemical reaction of converting the reactants into products. These turn equation (2) into

$$F = F_{CKT} = \frac{v_{max,f_E(T)E_tS}}{K+S} f_R(T), \tag{6}$$

where

$$v_{max} = v_{max,0} f_v(T), \tag{7}$$

$$K = \frac{v_{max}}{k_1^+} = K_0 f_K(T), \tag{8}$$

$$f_R(T) = 1 - exp\left(-\frac{\Delta G_R}{RT}\right), \tag{9}$$

and $v_{max,0}$ and $K_0$ are values of $v_{max}$ and $K$ evaluated at temperature $T_0$, respectively. Moreover, following the definition of $K$ in equation (2), we adopted the assumption that $v_{max}$ is much greater than $k_1^-$ in equation (8). In equation (6), $f_R(T)$ is computed by following Jin and Bethke (2003), with $\Delta G_R$ dependent on its value at standard conditions and the reaction quotient of the chemical reaction. However, except when there is significant product inhibition, $f_R$ may be set to one, which is adopted in the remainder of this paper. We next formulate $f_E(T)$, $f_v(T)$ and $f_K(T)$.

A two-state model is used to formulate the temperature dependent function $f_E(T)$ as

$$f_E(T) = \frac{1}{1 + exp\left(-\frac{\Delta G_E}{RT}\right)}, \tag{10}$$

with $R$ being the universal gas constant, and protein-unfolding Gibbs free energy

$$\Delta G_E = \Delta C_p \left[(T - T_H) - T ln\left(\frac{T}{T_S}\right)\right]. \tag{11}$$

Here $\Delta C_p$ is the heat capacity of protein unfolding, which is negative of the negative heat capacity of refolding used by Oliveberg et al. (1995) and is always positive due to proteins' hydrophobicity (Silverstein, 2020). $T_H$ is the temperature at which unfolding enthalpy is zero, and $T_S$ is the temperature at which unfolding entropy is zero. $\Delta C_p$, $T_H$ and $T_S$ are all functions of protein chain length (Ghosh and Dill, 2009), and, usually, $T_S$ is greater than $T_H$.

For $v_{max}$, applying the transition state theory (Eyring, 1935), we have

$$v_{max} = v_{max,0} f_v(T) = v_{max,0} \left(\frac{T}{T_0}\right) exp\left(-\frac{\Delta H_V}{RT}\left(1 - \frac{T}{T_0}\right)\right), \tag{12}$$

where $v_{max,0}$ is $v_{max}$ evaluated at reference temperature $T_0$, and $\Delta H_V$ is the enthalpy of activation and is temperature independent.

For the temperature dependence of $K$, applying the diffusion-limited chemical reaction model by von Smoluchowski (1917) indicates that $k_1^+$ is proportional to diffusivity. Therefore, by using the Stokes-Einstein equation of diffusivity (Miller,

1924), and considering the Arrhenius-type temperature dependence of dynamic viscosity of water, $k_1^+$ will have similar functional form of temperature dependence as $v_{max}$ (see (Tang et al., 2021) for more details), resulting in

$$f_K(T) = exp\left(-\frac{\Delta H_K}{RT}\left(1 - \frac{T}{T_0}\right)\right). \tag{13}$$

Combining equations (6)-(13), we have

$$F = F_{CKT} = v_{max,0}\frac{f_v(T)f_E(T)E_tS}{K_0f_K(T)+S}, \tag{14}$$

which describes the temperature dependence of the biochemical reaction rates in the absence of significant product inhibition.

In particular, by assuming that $S$ is much larger compared to $K_0f_K(T)$, we obtain

$$F_\infty = v_{max,0}E_tf_v(T)f_E(T) = r_0f_v(T)f_E(T), \tag{15}$$

which is the model proposed by Ratkowsky et al. (2005) to describe the temperature-dependent growth of various microorganisms (also see Corkrey et al., 2012; Corkrey et al., 2014).

*2.4 The relationship between chemical kinetics theory and MMRT*

In the analysis by (Hobbs et al., 2013), $k(T)$ was assumed to be equal to the reaction rates $F$ for enzyme assays with

$10 \times K_M$ at reference temperature (note that they denote $K$ with $K_M$). (They mentioned that enzyme assay data for barnase and its A43C/S80C mutant, which used substrate concentration of $2 \times K_M$, were corrected for possible $K_M$ dependence with $K_M$ determined at two temperatures.) This is equivalent to set $F_{MMRT}$ from equation (5) to be equal to $F_{CKT}$ in equation (14), which leads to

$$k(T) = v_{max,0}f_v(T)f_E(T)\frac{1+S/K_0}{f_K(T)+S/K_0}. \tag{16}$$

For the ease of parametric fitting (as will be described in section 2.5), taking the logarithm of equation (16) leads to

$$\ln k(T) = \ln v_{max,0} + \ln(1 + S/K_0) + \ln\left(\frac{f_v(T)f_E(T)}{f_K(T)+S/K_0}\right). \tag{17}$$

Equations (16) and (17) show that the temperature dependence of $k(T)$ is determined by $f_v(T)$, $f_E(T)$, $f_K(T)$, and the normalized substrate availability $S/K_0$. When enzyme assays are conducted with known values of $S/K_0$ and reference temperature (where $K_0$ is defined), the data can be used to derive the parameters $\Delta H_V$, $\Delta C_p$, $T_H$, $T_S$, and $\Delta H_K$. In our analysis,

since the activation enthalpy for the temperature dependence of the self-diffusion of water is almost a constant at 18 kJ mol$^{-1}$ (Mills, 1973), we set $\Delta H_K = \Delta H_V - 18$, so that only four parameters need to be derived from parametric fitting, which is one parameter more than required by MMRT.

Equation (16) or (17) can be used to analyze the motivating assumption and the two basic experimental assumptions that underlie MMRT. First, Hobbs et al. (2013) suggested that MMRT was motivated by noting that enzyme denaturation cannot satisfactorily explain the temperature dependence of catalysis rates. They then assumed that all enzymes are effectively in their active state to do catalysis, and attributed the decline in enzyme catalysis rate above an optimum temperature to the change of heat capacity associated with the enzyme catalysis. However, thermally reversible enzyme denaturation, as one type of enzyme denaturation, has been observed by many studies (Sizer, 1943;Alexandrov, 1964;Huang and Cabib, 1973;Maier et al., 1955;Weis, 1981), as well as by molecular dynamics simulations (McCully et al., 2008), and is ensured to occur by the ceaseless thermal motion of molecules and ions in the enzyme solution. Nonetheless, irreversible denaturation driven by heat does occur (Perdana et al., 2012), as it is necessary for the cooking of eggs or meat. Second, equations (16) and (17) clearly show that the enzyme assay-derived $k(T)$ is affected by substrate abundance, through the ratio ($S/K_0$) of substrate concentration $S$ to $K_0$. Third, because $K$ is temperature dependent, taking $S/K_0$ to be a relatively large value (e.g., 10 as adopted by Hobbs et al. (2013) and Peterson et al. (2004), hoping to "*minimize the effect of any possible increase in $K_M$ with temperature*") does not guarantee that the term $(1 + S/K_0)/(f_K(T) + S/K_0)$ reduces to a value of one (see section 3.1 for more details). Therefore, not only is the validity of their motivating presumption compromised, but also that of their experimental assumptions for deriving MMRT. Instead, as we will show later, a proper incorporation of thermally reversible enzyme denaturation is sufficient to explain the non-monotonic temperature dependence of enzyme catalysis rate, and substrate availability and the temperature dependence of $K$ can modulate the optimal temperature ($T_{opt}$) where biochemical reaction rate is maximized.

*2.5 Empirical data reanalysis*

We extracted the assay data of all seven enzymes from Hobbs et al. (2013) and all five enzymes from Peterson et al. (2004) to evaluate the validity of chemical kinetics theory. (We did not try to analyze data from soils, as that would involve a more comprehensive model, which is beyond the scope of this study.) Since we were not able to extract the reaction rates

directly from the figures in these studies, nor obtain the original data, we normalized rates for each enzyme with its own rate

at a selected reference temperature $T_r$, based on the criterion that the data point at $T_r$ is crossed by lines of their original

numerical fitting (where (Hobbs et al., 2013) used MMRT, while (Peterson et al., 2004) used their equilibrium model). In the

logarithm form (i.e., $\ln k(T)$), this normalization ensures that the values of $\ln k(T) - \ln k(T_r)$ used as observations at different

temperatures are independent of the value of $\ln v_{max,0} + \ln(1 + S/K_0)$ at the reference temperature $T_0$ of the enzyme essay.

We obtain the best fitting parameters by using the "fminsearch" function from MATLAB R2020b to minimize the summed

difference between modeled and measured values of $\ln k(T) - \ln k(T_r)$. Because we were not able to digitally extract

meaningful uncertainty of the observations from the figures either in Peterson et al. (2004) or in Hobbs et al. (2013), (which is

needed to compute the uncertainty using the Monte Carlo method), we do not compute uncertainties of the estimated

parameters. (We also tried using finite difference to approximate the Hessian matrix of the cost function at the best parameter

estimates obtained by "fminsearch". However, the ill-condition of the approximated Hessian matrix prevents us from

estimating the parametric uncertainty meaningfully. We did not try the bootstrapping method due to too few data points

available. However, the already excellent parametric fitting leads us to believe that if more data are available to derive the

uncertainty of the parameters, the conclusions will remain the same.)

### 3. Results

For the data of all twelve enzyme assays, the chemical kinetics theory obtained almost perfect model-data fitting with

the "fminsearch" computed best fit parameters (Figure 2). The $R^2$ values for the linear regression between model predictions

and observations are approaching 0.99 or 1.00 for 10 cases, and the lowest value is 0.85 for Barnase (Figure 2f). The best-fit

heat capacity $\Delta C_p$ of the thermodynamically reversible conversion between native and non-native conformations of the

enzymes are all positive, varying between 1.34 kJ mol$^{-1}$ K$^{-1}$ (for Adenosine deaminase in Figure 2i) and 22.74 kJ mol$^{-1}$ K$^{-1}$

(for Aryl-acylamidase in Figure 2k), in agreement with the range reported in Figure 2C by Ghosh and Dill (2009).

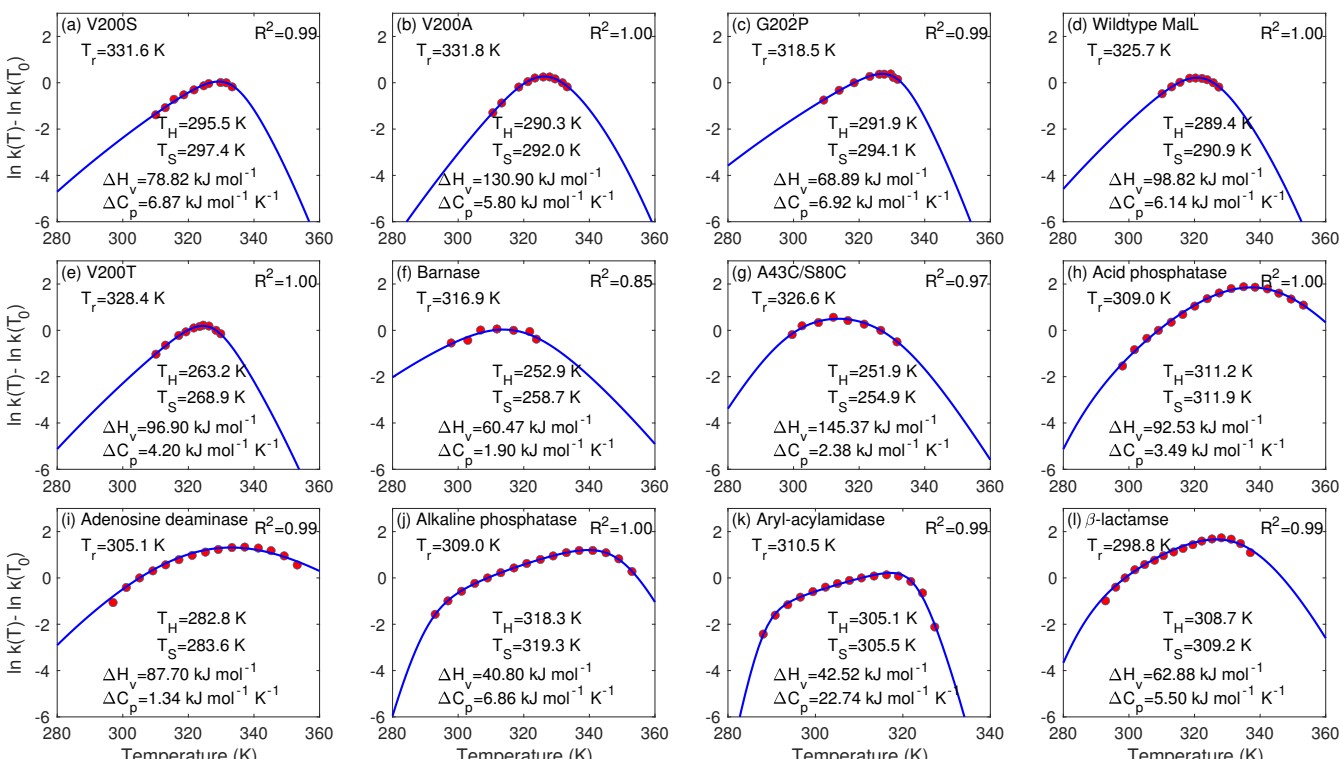

Figure 2. Fitting of the chemical kinetics theory (solid lines) to the enzyme assay data (red filled circles). Panels (a)-(g) are measurements from Hobbs et al. (2013) and panels (h)-(l) are from Peterson et al. (2004). $T_r$ is the reference temperature used in the data extracted from published figures (and it is different from the reference temperature $T_0$ that was actually involved in the enzyme assay experiments). $R^2$ is for the linear regression between the model predictions with best-fit parameters (blue lines) and measurements (in red circles). Following their original studies, parametric fitting for panels (f) and (g) used $S/K_0=2$, while others used $S/K_0=10$.

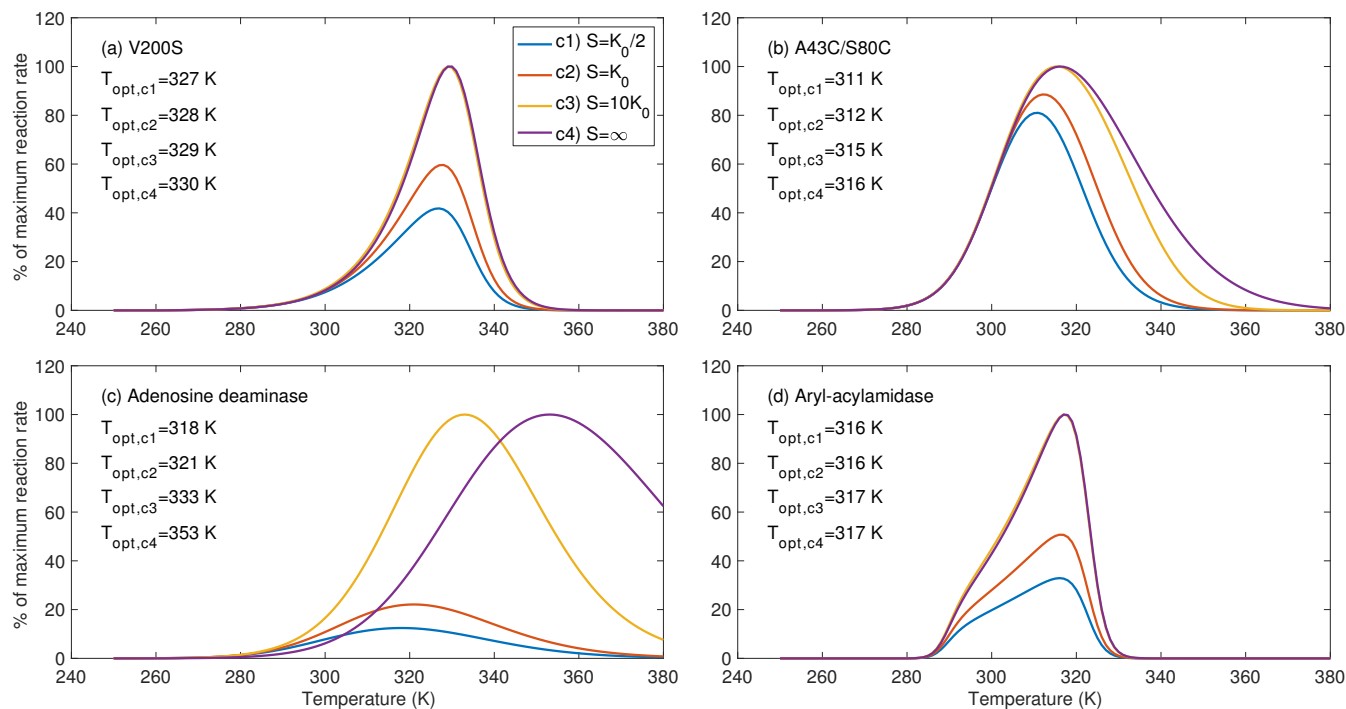

**Figure 3. Examples for the influence of substrate availability on the temperature response of enzymic reaction rates. For substrate level c4), the rate curve corresponds to $f_v(T)f_E(T)$ from equation (15).**

For the four example enzymes picked from the estimated parameters from Figure 2, we found the optimal temperature

(i.e., where the reaction rate reaches its maximum) has varying dependence on substrate availability (Figure 3). (We do not

show temperature response curves of the other eight enzymes, because they respond similarly to substrate availability, and

thus do not change our conclusions here.) All examples show that as substrate availability increases, the optimal temperature

increases and the temperature response curve shifts towards higher temperatures. For enzyme aryl-acylamidase, the actual

physiological optimal temperature under the saturating substrate concentration (i.e. when $S = \infty$, computed by equation (15))

equals the emergent optimal temperature at a substrate concentration of $10K_0$, and is 1 K higher than those at substrate

concentrations of $K_0$ and $K_0/2$ . For enzymes V200S and A43C/S80C, the optimal temperature at substrate concentration $10K_0$

is 1 K lower than the physiological optimal temperature (Figure 3a, b). However, this difference is 20 K for adenosine

deaminase (Figure 3c). These results clearly demonstrate that substrate availability plays a potentially important role in the

emergent temperature response of biochemical reaction rates. Nevertheless, we note that this prediction of substrate abundance

induced shift of optimal temperature should only be confronted with measurements of single enzyme reactions. As we will discuss later, the relationship between optimal temperature and substrate abundance in real soils is much more complicated.

**4. Discussion and conclusion**

Our theoretical analysis suggests that, even for a single-substrate-single-enzyme reaction, its temperature response involves contributions from at least four processes: (1) the thermally reversible transition between native and non-native state enzymes (which is ensured by the ceaseless thermal motions of molecules and ions in the enzyme solution); (2) the binding between native-state enzymes and substrates to form enzyme-substrate complexes; (3) the transition state activation of the enzyme-substrate complex; and (4) the thermodynamic feasibility for the biochemical reaction to generate product molecules. The chemical kinetics theory is able to explicitly account for all four processes, and can be extended to include more processes when more complex biochemical reactions are considered (e.g., as discussed in Tang et al. (2021)). In contrast, in spite of its simpler form, MMRT may have misinterpreted the functional relationship between measured biochemical rates and temperatures, in particular, by being unable to account for the modulation of optimal temperature and overall temperature response curve due to substrate availability. One consequence is that MMRT may be misinterpreting the inferred optimal temperature as the true physiologically optimal temperature (under the saturating substrate concentration), and regarding a measured shift of optimal temperature as evolutionary adaption. We find that higher optimal temperatures can be achieved under higher substrate availability. Further, the inferred temperature dependence by MMRT also includes contributions from the temperature sensitivity of affinity parameter, but the rate falloff at high temperature is not determined by the temperature sensitivity of the affinity parameter. Therefore, if one is using MMRT to represent reaction rate temperature dependencies and also includes a temperature-dependent substrate affinity parameter, the resultant model risks double counting the temperature response.

Recently, Numa et al. (2021) and Robinson et al. (2020) observed that adding plant litter or glucose to soil incubation samples resulted in lower optimum temperatures of soil respiration (when fitted with MMRT). As adding more substrate is most likely increasing the substrate concentration in soil, the lower optimum temperature appeared opposite to what the chemical kinetics theory predicts. However, we acknowledge that to apply the chemical kinetics theory to soils requires a model that considers the interactions between substrates, microbes and organo-mineral interactions. Since sorption interactions

between organic matter and soil minerals tend to increase the overall activation energy or enthalpy of carbon use by microbes (Tang and Riley, 2015), the newly added substrate most likely is having a lower activation energy than organic substrate that is already in soil, and will lead to a decrease in the optimal temperature of soil respiration. We demonstrated this with an example in Figure 4, where temperature response curves for a low and high activation energy cases are computed using equation (15), and found that lowering the activation energy reduced the optimal temperature by 2 K. Therefore, what Numa et al. (2021) and Robinson et al. (2020) observed could be well resulted from a shift in substrate type, which can be modelled through explicit representation of substrate competition and organo-mineral interactions (which was discussed in (Tang and Riley, 2013;Tang and Riley, 2015)).

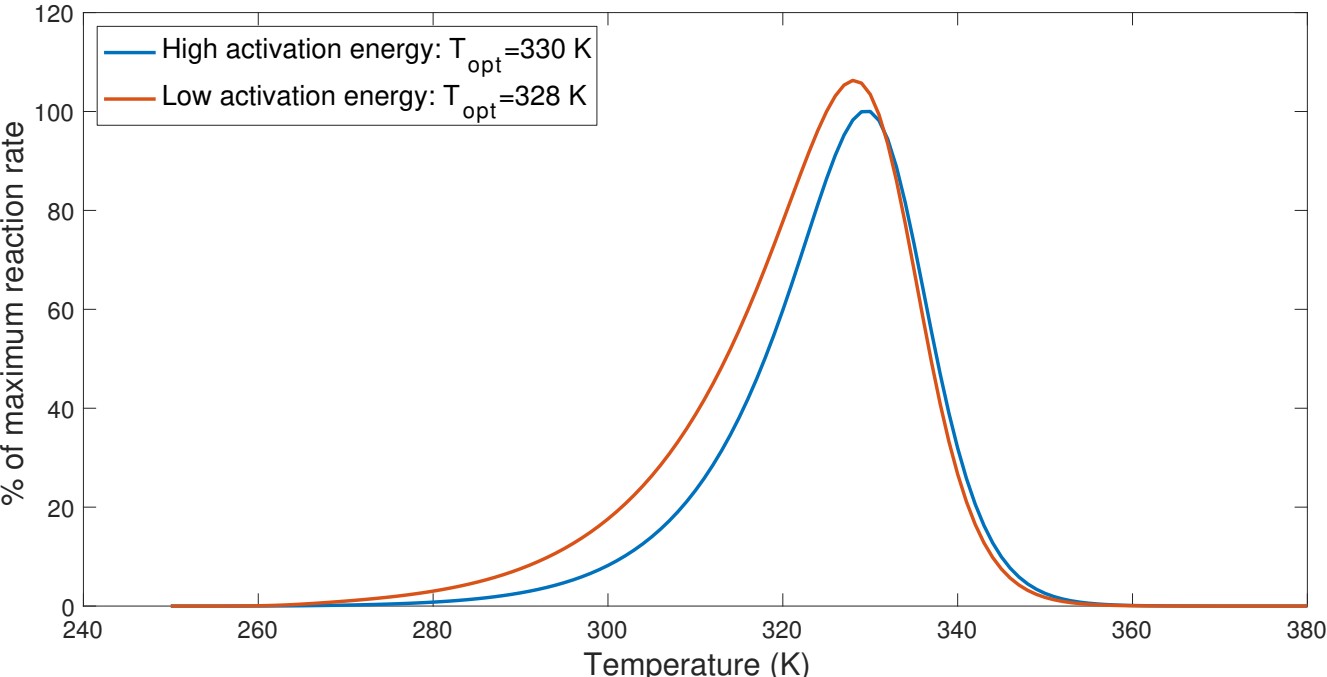

**Figure 4.** An example to show that lower activation energy will lead the optimal temperature to shift towards lower values. The curves are drawn based on equation **(15)**, with the high activation energy case using parameters from V200S, and the low activation energy case reduced $\Delta H_V$ from 78.82 kJ mol$^{-1}$ to 58.82 kJ mol$^{-1}$.

One outstanding difference between MMRT and the chemical kinetics theory is that MMRT generally infers a negative heat capacity (associated with the catalysis process), while the chemical kinetics theory infers a positive heat capacity of protein unfolding (associated with thermally reversible enzyme denaturation). Interestingly, if the heat capacity of protein refolding is substituted for the heat capacity of unfolding in the description of thermally reversible enzyme denaturation, its

sign becomes negative (Oliveberg et al., 1995). Some other studies that have used a similar framework of MMRT have associated the heat capacity with the enzyme-substrate binding process and have also found its value as being negative (e.g., Wang et al., 2009;Buczek and Horvath, 2006;Dullweber et al., 2001). However, using molecular dynamics simulations, Aqvist and Van der Ent (2022) inferred the heat capacity to be zero for both catalysis and binding processes for a designer enzyme 1A53-2.5. Moreover, Aqvist and Van der Ent (2022) and Aqvist (2022) suggested that the non-monotonic relationship between temperature and catalysis rate can be explained by the existence of an equilibrium between active enzyme substrate complex $E_nS$ and inactive enzyme substrate complex ($E'_nS$). To some extent, the conceptual model by Aqvist and Van der Ent (2022) is equivalent to the chemical kinetics theory, if the latter allows the inactive enzymes to form inactive enzyme-substrate complexes. We acknowledge that the finding of zero heat capacity for both catalysis and binding processes has been debated in Lear et al. (2023) and Aqvist (2023), but it is concluded that different kinetic models can fit the measured temperature dependent catalysis rates equally well. In particular, Aqvist (2023) noted that a kinetic model considering thermally reversible enzyme denaturation fits the observations equally well. However, deducing a non-zero heat capacity for both catalysis and binding processes seems to require one to ignore the thermally-reversible enzyme denaturation, which is ensured by the ceaseless thermal motion of molecules and ions in the enzyme solution.

Combining the transition state theory and the protein denaturation model by Lumry and Eyring (1954), Peterson et al. (2004) proposed an equilibrium model that includes both reversible and irreversible enzyme denaturation to explain their observed non-monotonic relationship between temperature and catalysis rates. However, because they assumed a constant enthalpy for the reversible enzyme denaturation, their Gibbs free energy of enzyme unfolding became a linear function of temperature. This linear function contrasts with the nonlinear function (i.e. equation (11)) and the existence of multiple native protein states that are usually observed in protein physics (Ghosh and Dill, 2009;Silverstein, 2020;Sheng and Pan, 2002;Finkelstein and Ptitsyn, 2016). Further, their model involves an explicit temporal dependence in the formulated catalysis rates, which introduces one more parameter (i.e., time) than the chemical kinetics model. Moreover, Peterson et al. (2004) also assumed their enzyme essays are substrate saturated, while we show above that such an assumption could very well be invalidated by the temperature dependence of substrate affinity parameter.

In summary, we contend that the chemical kinetics theory, by incorporating the observed thermally reversible transitions of enzymes between their native and non-native states (which occurs even in the absence of substrate molecules due to the ceaseless thermal motion of molecules and ions in the enzyme solution) (Anfinsen, 1973;Finkelstein and Ptitsyn, 2016;Sizer, 1943;Oliveberg et al., 1995), the diffusion-limited chemical reaction theory by (von Smoluchowski, 1917), and

325 the transition state theory by (Eyring, 1935), can satisfactorily explain the non-monotonic relationship between temperature and catalysis rates, and is a better mechanistic representation of the temperature dependence of enzyme-catalyzed biochemical rates than MMRT.

  Can chemical kinetics theory be upscaled to an organism from the single-substrate-single-enzyme examples presented here? While it is likely impossible to demonstrate such a scaling analytically, using the Ohm's law analogy from

330 Tang et al. (2021), where the temperature dependence of the emergent kinetic parameters (i.e., the overall $v_{max}$ and $K$) for chains of enzymes are found to follow similar forms as described by the chemical kinetics theory, we can qualitatively assert that the answer is true. Indeed, some previous studies (e.g., Ratkowsky et al., 2005;Corkrey et al., 2012;Ghosh et al., 2016) have showed that even equation (15) (which excludes substrate dependence) is able to satisfactorily describe temperature dependent growth of many organisms. Particularly, the success in capturing the temperature-dependent bacterial growth rate

in Ghosh et al. (2016), where they extended the thermally reversible enzyme denaturation in equation (15) to include all lethal proteins sampled from the proteome of mesophilic and thermophilic bacteria, suggesting that the chemical kinetics theory is very likely scalable. Therefore, all these successful applications imply that the chemical kinetics theory should have the potential to be applied to microbes, animals, and plants.

  Finally, because almost every microbe, animal, and plant is able to respire on multiple substrates (Madigan et al.,

2009;Cooper and Hausman, 2007) and the natural availability of those substrates usually fluctuate, the chemical kinetics theory and the equilibrium chemistry approximation kinetics for substrate competition networks (Tang and Riley, 2013) together suggest that a given organism will be unlikely to have either a fixed temperature response curve or optimal temperature even with a fixed proteome distributions. Rather, they both are likely to be dynamic.

## Acknowledgement

This research was supported by the Director, Office of Science, Office of Biological and Environmental Research of the US Department of Energy under contract no. DE-AC02-05CH11231 as part of the Belowground Biogeochemistry Science Focus Area and the Synthesis and Computation (RUBISCO) Scientific Focus Area. J.Y. Tang is also supported by the National Science Foundation under award number 2125069 and the Department of Energy, Office of Biological and Environmental Research, Genomic Sciences Program through the LLNL Microbes Persist Science Focus Area. Financial support does not constitute an endorsement by the Department of Energy of the views expressed in this study. The authors declare no conflicts of interest. We sincerely appreciate Prof. Louis Schipper and an anonymous reviewer, whose criticism and suggestions have help significantly improved our paper. We also thank Prof. Ken A. Dill for providing helpful insights on protein physics.

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
