# Peer review of "A chemical kinetics theory for interpreting the non-monotonic temperature dependence of enzymatic reactions"

_Biogeosciences, 2023_

## Author Comment (AC1)

**Response to reviewer 1.**

We appreciate Prof. Louis Schipper's careful assessment of our manuscript, which has helped us greatly improve its readability and scientific quality. We respond to his comments point by point below. It is noted that while addressing his comments, we also have taken the other reviewer's suggestions into account, so that a balanced decision is made on the requested changes to the manuscript. We italicize the corresponding changes in the manuscript for clarity.

**Comment 1**: This paper comprises two major elements (i) the paper starts with a critique of macromolecular rate theory (one of several models of the temperature response of enzyme activity) and (ii) development of chemical kinetics theory to predict temperature dependence of enzymatic reactions which includes the assumption of an equilibrium between active and reversibly denatured enzymes. It is important that we acknowledge that no 'simple' model will fully capture the complexity of individual enzymes and their coupled activity in biochemical pathways. There are hundreds/thousands of interacting atoms that play a role in enzyme catalysis and so we are attempting to develop much simpler models that capture the general behaviour of enzymes and biogeochemical processes in useful and predictive ways.

**Response**: We fully agree with the statement that no simple model can fully capture the complexity of individual enzymes and their coupled activity in biochemical pathways. Our attempt here is to address features missed in the macromolecular rate theory by considering the alternative chemical kinetics theory. In particular, we believe, as the developers of MMRT did, that a good theory to provide mechanistic insights should be able to incorporate as many observed major phenomena as possible, while being sufficiently simple and interpretable.

**Comment 2**: The authors present a model for the temperature dependence of enzyme rates that postulates an equilibrium between native and non-native conformations of the free enzyme (Figure 1b). This is a variation on the "equilibrium model" of (Daniel and Danson, 2013) that postulates an equilibrium between active and inactive conformations of the enzyme-substrate complex. This equilibrium model gave rise to macromolecular rate theory (MMRT) which captures the equilibrium in a heat capacity term. These three models are part of a large family of models that date back at least 70 years which try to rationalise the deviations from "Arrhenius behaviour" seen for enzymes, organisms and ecosystems and specifically capture the observed temperature optima. Thus, the current manuscript continues that tradition and testing this new model is a useful endeavour – in particular characterising the equilibrium via experiment and/or simulation.

**Response**: We agree that our chemical kinetics theory-based model is one kind of equilibrium model that tries to explain the deviation from "Arrhenius behaviour" observed for enzymes, organisms, etc. It is also our purpose to review the fundamental mechanisms responsible for observed temperature sensitivities and to compare and contrast how previous efforts have included these processes. We hope our manuscript will help readers to better understand existing

models that parameterize these deviations from "Arrhenius behaviour" and how they may be applied in their analyses.

**Comment 3**: The model presented in this manuscript is motivated (in the title, abstract and introduction) by two criticisms/assertions (ln 36 onwards) about MMRT which are incorrect: Firstly, MMRT does not assert that the Michealis-Menten equilibrium constant, $KM$, is independent of temperature. This would be to ignore the basic thermodynamic relationship between equilibrium and temperature: $\ln K = (-\Delta G/RT)$. Secondly, MMRT does not assume that the enzyme is saturated at all temperatures. MMRT models the temperature dependence of the rate constant, $k$, for rates at different scales (enzymes, organisms, ecosystems). For example, at high concentrations of substrate, the Michealis-Menten equation simplifies to rate = $k_{cat}$[E] and MMRT provides a model for the temperature dependence of $k_{cat}$. At low substrate concentrations the Michealis-Menten equation simplifies to rate = $(k_{cat}/K_M)$[E][S] = $k_2$[E][S] and MMRT provides a model for $k_2$ (which includes $K_M$). This is discussed in some detail by Arcus and Mulholland (2020). When modelling respiration rates in soil, for example, we expect that substrate concentrations are low and thus the latter simplification applies. We can nonetheless rule out $K_M$ as the source of deviations from Arrhenius behaviour for the following reason: If the non-Arrhenius behaviour for soil respiration were a consequence of $K_M$ then we would expect this behaviour to disappear at high substrate concentrations and this is not the case when glucose is added to soil. Indeed, the deviations from Arrhenius kinetics become more pronounced when glucose is added rather than less (e.g., Numa *et al.*, 2021). Thus, the deviations from Arrhenius behaviour, as modelled by MMRT, are not a function of the temperature-dependence of $KM$.

**Response**: We partially agree with this comment. Yes, it is true that MMRT does not assert that the Michealis-Menten equilibrium constant, $K_M$, is independent of temperature. Nor does MMRT assume that the enzyme is saturated at all temperatures. However, we noted that the two experimental studies used by Hobbs et al. (2013) to derive the parameters of MMRT and therefore the overall parametric fitting, did not consider the temperature dependence of $K_M$ or various levels of substrate concentrations. Specifically, Hobbs et al. (2013), in their description of "*Enzyme assays*", wrote that "*Temperature profiles were determined by measuring activity at 2−3 °C intervals using substrate at either 2 × $K_M$ (barnase) or 10 × $K_M$ (MalL). Barnase and A43C/S80C data were corrected for $V_{max}$ based on $K_M$ determinations at two temperatures.*" From this, we infer that the temperature dependence of $K_M$ is convolved with the estimated temperature dependence of $V_{max}$, and therefore may have led to uncertainties. Based on our new theory, we found that the temperature dependence of $K_M$ will lead to a substrate modulation of the emergent temperature dependence of catalysis rate (Figure 3), where at a given temperature for a given substrate, higher substrate concentration will increase the catalysis rate temperature sensitivity. In particular, we clarify that the chemical kinetics theory does not suggest that the $K_M$ temperature dependence will lead to the catalysis rate falloff as temperature increases over an

optimal, but it does modulate the value of optimal temperature for a given substrate through its relationship with substrate abundance (Figure 3).

**Comment 4**: While at least half of the manuscript is devoted to development of a chemical kinetics theory, this is not reflected in the title and only gets a single line in the abstract. To me, this paper is about the new model and both the title and abstract should reflect this approach. It is more than reasonable to debate the relative value of MMRT in the discussion when taking into account the general comment above.

**Response**: We agree and have revised the title to be "A reanalysis of the foundations of the macromolecular rate theory and an alternative based on chemical kinetics theory". We also revised the abstract accordingly:

"*The macromolecular rate theory (MMRT) has been proposed as a mechanistic scheme to describe the temperature dependence of enzymatic reactions, and has enjoyed quite some popularity recently. MMRT was motivated by assuming that enzyme denaturation is not sufficient to explain the decline of enzyme activity above an optimal temperature, and was derived with two experimental assumptions: (1) the half saturation parameter is independent of temperature; and (2) when the substrate concentration is kept at 10 times of the half saturation parameter at reference temperature, the enzyme assays are substrate saturated under all experimental temperatures. We show that thermally reversible enzyme denaturation could be essential to consistently interpret the temperature dependence of enzymatic reactions, and due to the temperature-dependence of the half saturation parameter, neither of the experimental assumptions of MMRT held. Consequently, the MMRT estimated temperature sensitivity of the maximum catalysis rate is inaccurate. It can mischaracterize temperature-related biochemical behaviors, such as inferring the existence of a unique optimal temperature where biochemical rate peaks, and the shift of this optimal temperature as an indicator of thermal acclimation or adaptation. We proposed a chemical kinetics theory that explicitly incorporates the observed thermally reversible enzyme denaturation, von Smoluchowski's diffusion-limited chemical reaction theory, and Eyring's transition state theory to interpret the temperature dependence of enzymatic reactions. Since the chemical kinetics theory performed equally successful in fitting the enzyme assay data used in deriving MMRT, and has incorporated more relevant empirical observations and well-established theories than MMRT, we recommend it as a better candidate for mechanistic modeling of the temperature dependence of biogeochemical rates. However, MMRT is still a better model than the conventional $Q_{10}$ and Arrhenius functions for describing the emergent temperature dependence of biochemical rates.*"

**Comment 5**: In the introduction, not only should MMRT be described and critiqued as a temperature dependence model but the array of other equilibrium models, and (e.g. the DAMM model) (Davidson *et al.*, 2012), square root model etc…

**Response**: We now expanded the introduction per this suggestion and a similar suggestion raised by the other reviewer, so that a more comprehensive landscape is painted for the progress on modeling the deviation from "Arrhenius equation". Particularly, we added a paragraph to discuss some important historical developments of the equilibrium model, and highlighted the commonality of thermally reversible enzyme denaturation.

*"Besides MMRT, a few other models with mechanistically interpretable parameters are also capable of equally well interpreting the non-monotonic temperature dependence of enzyme modulated reactions, including growth rates. Notably, Sharpe and Demichele (1977) proposed a model that incorporates the empirical observation of thermally reversible enzyme denaturation and the transition state theory (Eyring, 1935). Specifically, they considered that enzymes are in reversible transition between three states, one cold-induced inactive state, one heat-induced inactive state, and one active state which is able to carry out the catalysis. By assuming reactions to be substrate unlimited, they obtained a model with five thermodynamic parameters that is able to almost perfectly fit published temperature dependent growth rates of eight poikilothermic organisms (see their Figures 5 and 6). (The applicability of the Sharpe-Demichele model to growth rates of an organism is based on the assumed existence of control or master enzymes (Johnson and Lewin, 1946).) Motivated by the success of Sharpe and Demichele (1977) and the work on thermally reversible protein denaturation by Murphy et al. (1990), Ratkowsky et al. (2005) grouped the two inactive states into one, and, again assuming no-substrate limitation, derived a model with two thermodynamic parameters and two enzyme informatic parameters, which was able to very accurately fit 35 sets of observed temperature dependent bacterial growth rates. The model by Ratkowsky et al. (2005) was later used by (Corkrey et al., 2012) and (Corkrey et al., 2014) to successfully interpret the temperature dependent growth rates of many more poikilothermic organisms. Ghosh et al. (2016) extended the model by Ratkowsky et al. (2005) to include the thermally reversible denaturation of many enzymes and proteins informed by proteomics, and were able to satisfactorily interpret the measured temperature-dependent growth rates of mesophiles and thermophiles.*

*The thermally-reversible enzyme denaturation occurs due to the thermal motion of molecules and ions in the solution of enzyme proteins (Finkelstein and Ptitsyn, 2016). As thermal motion is ceaseless, according to Boltzmann's law in statistical mechanics (Feynman et al., 2011), enzyme molecules will be distributed among different configurations that can be quantified by their respective energy status. Therefore, even under room temperature, some of the enzyme molecules will not be in their biologically active native states. That is, at any given temperature, only a fraction of enzymes is able to catalyze the corresponding biochemical reaction. Consequently, by not explicitly taking into account the thermally-reversible enzyme denaturation (or by assuming all enzyme denaturation are irreversible), we believe MMRT may have missed some important mechanistic insights on the temperature control of enzymatic reactions."*

**Comment** 6: The developed theory is only tested on data from single enzymes (Hobbs et al 2013) and not biogeochemical processes so how this work translates to a journal that focuses on biogeochemical modelling is not clear. Would this be better submitted to a biochemistry journal that focusses on enzymes? Also, see line 149 the authors state a more complex model is needed for soil, so it is not clear how the authors suggest a developed model fits in biogeochemistry.

**Response**: Since our purpose is to reanalyze MMRT and propose an alternative, and MMRT is built upon the data that are used in this manuscript, we believe that it is reasonable to leave further development in future studies. Additionally, because MMRT is now popularly used for biogeochemical modeling or analysis, we believe that publishing our manuscript with biogeoscience can help the research community to improve the understanding and modeling of various relevant biochemical processes.

We noted that Ghosh et al. (2016) have successfully applied the Ratkowsky et al. (2005) model to include many lethal proteins sampled from proteomes of some mesophilic and thermophilic bacteria. Since the Ratkowsky model is a special case of the chemical kinetics theory, these previous studies by Ghosh et al. (2016), Ratkowsky et al. (2005), and Corkrey et al. (2012) together thus show that the chemical kinetics theory here is scalable from single enzyme to unicellular, and multicellular organisms.

Finally, in order to further show the validity of the chemical kinetics theory, we show that the observed reduction in optimal temperature upon addition of plant litter or glucose by (Robinson *et al.*, 2020; Numa *et al.*, 2021) can potentially be explained as a shift in substrate type. Specifically, in the newly added Figure 4, which is also shown below (Figure S1), we argue that since organo-mineral interaction tends to increase the activation energy of existing SOM substrates, our theory predicts that the lower activation energy of newly added plant litter or glucose will lower the emergent temperature optimal. Thus, we believe a strong case to publish this manuscript with biogeosciences is presented.

[Figure]

Figure S1. Lowering the activation energy will shift the optimal temperature towards lower values. This is shown as Figure 4 in the revised manuscript.

**Comment 7**: Ln 42 the authors should define what reversible denaturation is in their framework. As large molecules, enzymes constantly undergo large fluctuations resulting in an array of conformations that bind the substrate to varying degrees. Are the authors assuming only 2 states (active and inactive) rather than a continuum of binding/catalytic states with varying Kms?

**Response**: We simply define reversible denaturation as the transition between native states (that are able to bind the substrate and catalyze the reaction) and non-native states (that are not able to bind the substrate and catalyze the reaction). This is simpler than considering more than two states, but is sufficient to account for the observed (1) existence of thermally reversible enzyme denaturation, and (2) the non-monotonic temperature dependence of enzymatic reactions. In the revised manuscript, we now defined the reversible denaturation as "the dynamic transition between its native folded state and the unfolded state as a function of temperature and solution conditions (e.g., Oliveberg et al., 1995;Anfinsen, 1973)", and made a more explicit description about how thermally reversible enzyme denaturation was used in several existing models by adding a new paragraph in the introduction. We have also consulted Prof. Ken A. Dill, an expert in protein physics, to make sure this definition is simple but sufficient. (Also see response to comment 5.)

**Comment 8**: Ln 90 Schipper and Liang and others primarily suggested using MMRT rather than Q10 or Arrhenius functions because neither latter function resulted in a measure of a Topt (and therefore Tinf).

**Response**: We agree with this point, and acknowledge that MMRT improved on the Q10 and Arrhenius functions in this regard. However, we also acknowledge that there are alternative models that can equally well capture the non-monotonic relationship between biochemical rates and temperature. Also, per the suggestion from the other reviewer, we expanded the introduction to make it more inclusive (also see response to 5).

**Comment 9**: Ln 205 and elsewhere (ln 79). There needs to be a clear description of the differences between the dCp for the different models. In the current paper, dCp is the change in heat capacity between the reversibly denatured state and the active state. In MMRT, the dCp is argued to be the heat capacity difference between the enzyme-substrate complex and the enzyme transition state. There is no reason that these should be the same size of sign as these are not the same processes.

**Response**: We agree that this issue is important to clarify. In particular, we dedicated the following paragraph to address these differences:

"*One outstanding difference between MMRT and the chemical kinetics theory is that MMRT generally infers a negative heat capacity (associated with the catalysis process), while the chemical kinetics theory infers a positive heat capacity of protein unfolding (associated with*

*thermally reversible enzyme denaturation). Interestingly, if the heat capacity of protein refolding is substituted for the heat capacity of unfolding in the description of thermally reversible enzyme denaturation, its sign becomes negative (Oliveberg et al., 1995). Some other studies that have used a similar framework of MMRT have associated the heat capacity with the enzyme-substrate binding process and have also found its value as being negative (e.g., Wang et al., 2009;Buczek and Horvath, 2006;Dullweber et al., 2001). However, using molecular dynamics simulations, Aqvist and Van der Ent (2022) inferred the heat capacity to be zero for both catalysis and binding processes for a designer enzyme 1A53-2.5. Moreover, Aqvist and Van der Ent (2022) and Aqvist (2022) suggested that the non-monotonic relationship between temperature and catalysis rate can be explained by the existence of an equilibrium between active enzyme substrate complex and inactive enzyme substrate complex. To some extent, the conceptual model by Aqvist and Van der Ent (2022) is equivalent to the chemical kinetics theory, if the latter allows the inactive enzymes to form inactive enzyme-substrate complexes. We acknowledge that the finding of zero heat capacity for both catalysis and binding processes has been debated in Lear et al. (2023) and Aqvist (2023), but it is concluded that different kinetic models can fit the measured temperature dependent catalysis rates equally well. In particular, Aqvist (2023) noted that a kinetic model considering thermally reversible enzyme denaturation fits the observations equally well.*"

**Comment 10**: Ln 133 enzyme denaturation was shown experimentally to have Topt without enzyme denaturation in Hobbs et al 2013. Are the authors invoking reversible denaturation here? See also ln 143 are the authors arguing for including irreversible denaturation or reversible denaturation? These are very different concepts.

**Response**: Hobbs et al. (2013) stated that enzyme denaturation cannot satisfactorily explain the observed Topt, so they propose MMRT which does not require enzyme denaturation. By attributing the observed Topt to changes in heat capacity associated with enzyme catalysis, MMRT effectively assumes that all enzymes are in their native states and are capable of forming enzyme-substrate complexes. In contrast, our new approach is based on well documented observations of thermally-reversible enzyme denaturation, and that models incorporating this reversible denaturation are able to describe the observed nonmonotonic temperature dependence of enzymatic reactions (e.g. the model by Ratkowsky et al.(2005)). We now revised this part of the text as:

"*Equation (16) or (17) can be used to analyze the motivating assumption and the two basic experimental assumptions that underlie MMRT. First, Hobbs et al. (2013) suggested that MMRT was motivated by noting that enzyme denaturation cannot satisfactorily explain the temperature dependence of catalysis rates. They then assumed that all enzymes are effectively in their active state to do catalysis, and attributed the decline in enzyme catalysis rate above an optimum temperature to the change of heat capacity associated with the enzyme catalysis. However, thermally reversible enzyme denaturation, as one type of enzyme denaturation, has been*

*observed by many studies (Sizer, 1943;Alexandrov, 1964;Huang and Cabib, 1973;Maier et al., 1955;Weis, 1981), as well as by molecular dynamics simulations (McCully et al., 2008), and is ensured to occur by the thermal motion of molecules and ions in the enzyme solution. Nonetheless, irreversible denaturation driven by heat does occur (Perdana et al., 2012), as it is necessary for the cooking of eggs or meat. Second, equations (16) and (17) clearly show that the enzyme assay-derived $k(T)$ is affected by substrate abundance, through the ratio $(S/K_0)$ of substrate concentration S to $K_0$. Third, because K is temperature dependent, taking $(S/K_0)$ to be a relatively large value (e.g. 10 as adopted by Hobbs et al. (2013) and Peterson et al. (2004), hoping to "minimize the effect of any possible increase in $K_M$ with temperature") does not guarantee that the term $(1+S/K_0)/(f_K(T)+S/K_0)$ reduces to a value of one (see section 3.1 for more details). Therefore, not only is the validity of their motivating presumption compromised, but also that of their experimental assumptions for deriving MMRT. Instead, as we will show later, a proper incorporation of thermally reversible enzyme denaturation is sufficient to explain the non-monotonic temperature dependence of enzyme catalysis rate, and substrate availability and the temperature dependence of K can modulate the optimal temperature ($T_{opt}$) where biochemical reaction rate is maximized."*

**Comment 11**: Ln 155 why can the uncertainty ranges of the fitted parameters not be determined? This should be about the fit of the data and not the variation in data extraction?

**Response**: Because that the parameter fitting problem is close to singular, we were not able to derive the uncertainty from the Hessian matrix using finite difference approximation. An alternative approach is to use Monte Carlo method with perturbed observations, but we were not able to meaningfully extract the uncertainty of observations from the log plots in Hobbs et al. (2013) and Peterson et al. (2004). However, we believe this shortcoming will not change the conclusion of our manuscript.

**Comment 12**: Ln 180, Figure 3 Is this modelled data? The authors argue that Topt increases with decreasing substrate concentration but do not provide error bounds on Topt for substrate concentrations ranging between Ko and infinite substrate. Is this a real change in Topt, that is, do the authors believe that a modelled increase in Topt of 1C (figure 3d) or 3C (Figure 3a) is real, based on a 4-parameter fitted model? This is something that could be experimentally tested. In contrast, measurement of respiration when glucose or litter was added to soil shows decreases the temperature optima (Robinson *et al.*, 2020; Numa *et al.*, 2021)

**Response**: Yes, Figure 3 presents model predictions using some of the parameters derived in Figure 2 (so there is no uncertainty). For measurements of temperature dependence from real soils, the situation is more complicated due to the involvement of many microbes, enzymes, substrates and mineral particles, as the reviewer mentioned previously. We note that our predictions should be confronted with measurements based on single enzyme catalyzed reactions

in solution, and have indicated this point in the revised text. For real soil systems, our previous work (Tang and Riley (2013, 2015)) shows that soil minerals will effectively increase the affinity parameter. Therefore, adding labile substrates can trigger a respiratory reaction that has lower affinity parameter and activation energy compared to the stabilized substrate already in soil. These interactions will cause a shift of the temperature optimal towards smaller values, as observed by the two studies quoted above. We acknowledge these complexities for real soils in the revised text, and note that the chemical kinetics theory needs to be implemented in a more comprehensive model to explain the observations by Robinson et al. (2020) and Numa et al. (2022). We also explained these issues in our response to comment 5.

**Comment 13**: Figure 3b why is the maximal rate of c3 (substrate 10x Ko) so much lower than when substrate is limiting c1. Would you not expect higher rates when substrate was not limiting?

**Response**: Thanks for catching this, we double checked the matlab script and identified a bug when plotting the results. Now the figures are corrected, but the derived numbers and conclusions remained the same.

**Comment 14**: Also figure 3d seems to have bulged in the curve at around 300 K, which is not normally seen in experimental data. Is this considered as realistic or a function of the model?

**Response**: The bulge was derived from the pattern in Figure 2k for Aryl-acylamidase, whose data are derived from Peterson et al. (2004). Their original data were also not very smooth. However, it was also caused by the bug in the plotting script. The corrected Figure 3d is smoother.

**Comment 15**: Ln 187 what is meant by the physiological optimum - what concentration of substrate is this referring to?

**Response**: We clarified that it refers to the optimal temperature at unlimited substrate concentrations as computed by equation (15).

**Comment 16**: Ln 197 MMRT has been used to capture differences due to changes in substrate availability on Topt in soil (Numa *et al.*, 2021).

**Response:** As we explained in the response to comments 6 and 12. We hypothesize that Numa et al. (2021) may have observed a competition effect on Topt associated with the change in substrate type. Our chemical kinetics theory when integrated with a complete model for SOM dynamics will be able to resolve this pattern, as we suggested in a prototype model presented in Tang and Riley (2015)).

**Comment 17**: Ln 201 Is the assertion that higher Topt occurs with high substrate availability is a consequence of the developed model but not observations? Should be clearly stated. Is this extrapolating beyond the parameters used to model the enzyme in the first place?

**Response**: It is a prediction based on the chemical kinetics theory. We clarified this point in the revision by adding a discussion on observations by Numa et al. (2021) and Robinson et al. (2020). Please also see responses to comments 6 and 12.

**Comment 18**: Ln 206 a comparison between dCp between the model developed by the authors and MMRT is not reasonable these are different phenomena as pointed out above. In the current paper, dCp is the change in heat capacity between the reversibly denatured state and the active state. In MMRT, the dCp reported is argued to be the heat capacity difference between the enzyme-substrate complex and the enzyme transition state. The dCp for transition state binding (that is MMRT dCp) for MTAP has been measured by Firestone et al (2017) and is same as inferred by MMRT fits within 5%.

**Response**: We now clarified the conceptual difference between these two heat capacities, to ensure readers are aware of the different definitions. Also see response to comment 9.

**Comment 19**: Ln 215 Can the authors formalise what was considered wrong with the La Mer study rather than broad statement that was considered flawed. This paper is not referenced in the early MMRT papers (Hobbs *et al.*, 2013) (Arcus *et al.*, 2016)

**Response**: One reviewer (who decided to be anonymous but may have been involved in the original author list of MMRT papers) of the earlier draft of this manuscript suggested that MMRT was partially motivated by the study La Mer (1933). In that paper, La Mer suggested that the rate constant should be $k_f = Z_f \cdot exp(-\Delta G_f/RT)$ for forward reaction, and $k_b = Z_b \cdot exp(-\Delta G_b/RT)$ for backward reaction, so that the equilibrium constant is $K_e = k_f/k_b = Z_f/Z_b \cdot exp(-\Delta G^o/RT)$, with $\Delta G^o = \Delta G_f - \Delta G_b$ (being a linear function of temperature). Because La Mer (1933) assumed a non-zero heat capacity for the free energy of activation, one will find $Z_f \neq Z_b$. However, $Z_f = Z_b$ is needed to make $K_e = exp(-\Delta G^o/RT)$, so that the theory is consistent with the transition state theory and thermodynamics. However, since the original MMRT papers did not cite La Mer (1933), and because that paper has subsequently been shown to be flawed, we removed discussion of it in our revised manuscript.

**Comment 20**: Ln 221. How can the authors argue that the developed model is a "better mechanistic representation of temperature dependent biochemical rates than MMRT"? This inference is based on a better fit to data using a model with 4 parameters (1 more than MMRT, as noted ln 140, so not surprising that the fit is better). There is no experimental testing of the

underlying mechanism proposed in their model. There is also no accounting for the addition of another parameter using, for example, AIC, which probably should be done.

**Response**: Our argument for the better mechanistic representation of our new model is that it is built upon its equally good parametric fitting (as compared to MMRT), inclusion of more observed phenomena (e.g., the existence of thermally reversible enzyme denaturation), consistent incorporation of the transition state theory and von Smoluchowski's chemical reaction theory, and indirect support from the molecular dynamics simulations. These make it worthwhile to introduce one more physically meaningful parameter. In particular, we are not proposing an alternative model based on statistical regression, rather that the value of our alternative approach is that it is based on a solid theoretical foundation of chemical kinetics and thermodynamics, which themselves are built upon a wide range of empirical observations. To address the reviewer comment and clarify these points, we have added text to the revised manuscript:

"*In summary, we contend that the chemical kinetics theory, by incorporating the observed thermally reversible transitions of enzymes between their native and non-native states (which occurs even in the absence of substrate molecules due to the ceaseless thermal motion of molecules and ions in the enzyme solution) (Anfinsen, 1973;Finkelstein and Ptitsyn, 2016;Sizer, 1943;Oliveberg et al., 1995), the diffusion-limited chemical reaction theory by (von Smoluchowski, 1917), and the transition state theory by (Eyring, 1935), can satisfactorily explain the non-monotonic relationship between temperature and catalysis rates, and is a better mechanistic representation of the temperature dependence of enzyme-catalyzed biochemical rates than MMRT.*"

**Comment 21**: Ln 229. Again the authors argue their model is "better" than the Peterson model simply because the Peterson model has an extra parameter and no AIC calculated between comparative models. It is unclear how the authors are considering the relative merits of different models. Is it better to consider the number of fitted parameters to describe the response or mechanistic representation?

**Response**: We noted that the advantage of the Peterson model also involves an inconsistency between their theory and support from empirical experiments, and have used a Gibbs free energy of unfolding that contradicts what is widely known in protein physics. In the revised manuscript, we have clearly spell out the issues associated with the Peterson model:

"*Combining the transition state theory and the protein denaturation model by Lumry and Eyring (1954), Peterson et al. (2004) proposed an equilibrium model that includes both reversible and irreversible enzyme denaturation to explain their observed non-monotonic relationship between temperature and catalysis rates. However, because they assumed a constant enthalpy for the reversible enzyme denaturation, their Gibbs free energy of enzyme unfolding became a linear function of temperature. This linear function contrasts with the nonlinear function (i.e. equation (11)) and the existence of multiple native protein states that are usually observed in protein*

*physics (Ghosh and Dill, 2009;Silverstein, 2020;Sheng and Pan, 2002;Finkelstein and Ptitsyn, 2016). Further, their model involves an explicit temporal dependence in the formulated catalysis rates, which introduces one more parameter (i.e., time) than the chemical kinetics model. Moreover, (Peterson et al., 2004) also assumed their enzyme essays are substrate saturated, while we show above that such an assumption could very well be invalidated by the temperature dependence of substrate affinity parameter.*"

In addition, we now stated more clearly how we evaluated the merits of a model:

*"In summary, we contend that the chemical kinetics theory, by incorporating the observed thermally reversible transitions of enzymes between their native and non-native states (even in the absence of substrate molecules) (Anfinsen, 1973;Finkelstein and Ptitsyn, 2016;Sizer, 1943;Oliveberg et al., 1995), the diffusion-limited chemical reaction theory by (von Smoluchowski, 1917), and the transition state theory by (Eyring, 1935), can satisfactorily explain the non-monotonic relationship between temperature and catalysis rates, and is a better mechanistic representation of the temperature dependence of enzyme-catalyzed biochemical rates than MMRT. "*

**Comment 22**: Overall, the current manuscript presents an interesting approach on the temperature dependence of enzyme-driven rates and investigation of the phenomena contained in the model warrants investigation. Our view is that this is sufficient in itself, and does not need to be prefaced by incorrect assertions about MMRT. In biogeosciences, we are seeking tradeoffs between fully mechanistic models of temperature response and those that are useful. As the saying goes: All models are wrong, some are useful.

**Response**:  We thank the reviewer for his constructive comments. We fully agree with the statistician George E.P. Box's sentiment that all models are wrong, some are useful. With that understanding and the reviewer's comments, we have rewritten the overall tone and goal of our manuscript to be a discussion of how our new approach can integrate well-established theory (e.g., law of mass action, thermodynamics, chemical kinetics theory), previous efforts (e.g., MMRT, Peterson, …), and observed phenomena (e.g., the widely observed thermally reversible enzyme denaturation).

**References**

Arcus, V.L., Prentice, E., Hobbs, J.K., Mulholland, A.J., Vander Kamp, M.W., Pudney, C.R., Parker, E.J., Schipper, L.A., 2016. On the temperature dependence of enzyme-catalysed rates. Biochemistry 55, 1681-1688.

Arcus, V. L. & Mulholland, A. J. Temperature, Dynamics, and Enzyme-Catalyzed Reaction Rates. Annual review of biophysics (2020) doi:10.1146/annurev-biophys-121219-081520.

Davidson, E.A., Samanta, S., Caramori, S.S., Savage, K., 2012. The Dual Arrhenius and Michaelis–Menten kinetics model for decomposition of soil organic matter at hourly to seasonal time scales. Global Change Biology 18, 371-384

Firestone, R. S., Cameron, S. A., Karp, J. M., Arcus, V. L. & Schramm, V. L. Heat Capacity Changes for Transition-State Analogue Binding and Catalysis with Human 5'-Methylthioadenosine Phosphorylase. ACS Chemical Biology 12, 464–473 (2017).

Hobbs, J.K., Jiao, W., Easter, A.D., Parker, E.J., Schipper, L.A., Arcus, V.L., 2013. Change in heat capacity for enzyme catalysis determines temperature dependence of enzyme catalyzed rates. ACS Chemical Biology 8, 2388-2393.

Numa, K.B., Robinson, J.M., Arcus, V.L., Schipper, L.A., 2021. Separating the temperature response of soil respiration derived from soil organic matter and added labile carbon compounds. Geoderma 400, 115128.

Robinson, J.M., Barker, S.L.L., Arcus, V.L., McNally, S.R., Schipper, L.A., 2020. Contrasting temperature responses of soil respiration derived from soil organic matter and added plant litter. Biogeochemistry 150, 45-59.

---

## Author Comment (AC2)

**Response to Reviewer 2**

We appreciate the reviewer's careful assessment of our manuscript. We respond to the comments point by point below. It is noted that while addressing his comments, we also had taken the other review's suggestions into account, so that a balanced decision is made on the requested changes to the manuscript. We highlight the corresponding changes in the manuscript by quoting them in italic font.

**Comment 1**: The paper provides a critique of the MMRT model, concentrating on its underlying assumptions as stated by the authors. This makes for a useful contribution since MMRT has proved popular in several fields, such as soil science. Should the model be flawed or incomplete then these applications may have reduced value, particularly in the journal's subject area. The paper does not make a direct comparison with alternative models, though it refers to a number of them, collectively termed `chemical kinetics theory'. Certainly, empirical models would not be of interest here, but in any case, I do not consider such comparisons are needed since they can needlessly burden a paper with material that the interested reader can seek out.

**Response**: We appreciate the reviewer's sentiment on our paper, which fully captured our intent for the design and content of this manuscript. We followed the reviewer's suggestions as much as we can, and believe that the revised manuscript is now much more helpful to readers of biogeosciences.

**Comment 2**: In the introduction there is a brief coverage of publications that refer to MMRT. I found this could have been more comprehensive. It could have dealt with a number of alternative models and, as noted earlier, that would burden the paper.

**Response**: We now expand the introduction of MMRT, and also the alternative mechanistic models. We now included the comprehensive review by (Grimaud et al., 2017) and (Noll et al., 2020), and added the following paragraph

*"Besides MMRT, a few other models with mechanistically interpretable parameters are also capable of equally well interpreting the non-monotonic temperature dependence of enzyme modulated reactions, including growth rates. Notably, Sharpe and Demichele (1977) proposed a model that incorporates the empirical observation of thermally reversible enzyme denaturation and the transition state theory (Eyring, 1935). Specifically, they considered that enzymes are in reversible transition between three states, one cold-induced inactive state, one heat-induced inactive state, and one active state which is able to carry out the catalysis. By assuming reactions to be substrate unlimited, they obtained a model with five thermodynamic parameters that is able to almost perfectly fit published temperature dependent growth rates of eight poikilothermic organisms (see their Figures 5 and 6). (The applicability of the Sharpe-Demichele model to growth rates of an organism is based on the assumed existence of control or master enzymes (Johnson and Lewin, 1946).) Motivated by the success of Sharpe and Demichele (1977) and the work on thermally reversible protein denaturation by Murphy et al. (1990), Ratkowsky et al. (2005) grouped the two inactive states into one, and, again assuming no-*

*substrate limitation, derived a model with two thermodynamic parameters and two enzyme informatic parameters, which was able to very accurately fit 35 sets of observed temperature dependent bacterial growth rates. The model by Ratkowsky et al. (2005) was later used by (Corkrey et al., 2012) and (Corkrey et al., 2014) to successfully interpret the temperature dependent growth rates of many more poikilothermic organisms. Ghosh et al. (2016) extended the model by Ratkowsky et al. (2005) to include the thermally reversible denaturation of many enzymes and proteins informed by proteomics, and were able to satisfactorily interpret the measured temperature-dependent growth rates of mesophiles and thermophiles. "*

**Comment 3**: L20-25. The authors note that Hobbs et al. (2013), Schipper et al. (2014) claim MMRT is able to better than the Arrhenius-like functions for various ecological properties. I suggest that it may be worth adding here that while Schipper et al. (2014) compares MMRT to an Arrhenius model it does not do any more than this, in that it does not, for example, choose to compare to the Ratkowksy 2005, Corkrey 2012 models.

**Response**: We now acknowledge that models, e.g. by Sharpe and DeMichele (1977), Ratkowsky et al. (2005), Corkrey et al. (2012) can be equally well in terms fitting the pattern. We also acknowledged how the MMRT papers mis-criticized the Ratkowsky model. In particular, the important study by Oliveberg et al. (1995) who reported negative heat capacity of protein refolding was actually supporting the Ratkowsky model and the chemical kinetics theory here.

**Comment 4**: L25-30. Similarly, I would add that Alster et al (1016) does the same thing, but in addition, I would add that those authors unnecessarily dismiss other models as 'empirical' when, on examination, they do not appear be (e.g. Corkrey et al (2012), Peterson et al (2004), Daniel & Danson (2013)). This is a point of irritation since such claims are too easily repeated.

**Response**: We added this more balanced view throughout the manuscript. Also see response to comment 2.

**Comment 5**: L30-35. Examples (e.g., Ratkowsky et al (1983)) of papers as having parameters that are not biologically interpretable. This seems a little unfair since that particular model is explicitly empirical. I suggest adding in references to other models that do have interpretable parameters and/or are based on thermodynamic principles such as those listed above.

**Response**: We added references of other models, and annotated them fairly.

**Comment 6**: Section 2.3. I found this derivation somewhat hard to follow (though interesting). Perhaps a little more could be added to assist non-specialists?

**Response**: We added more verbal descriptions to mathematical derivations, so that they are easier to follow.

**Comment 7**: L130-135. I stumbled over the wording 'The motivating assumption ... many observations'. Perhaps tweak this to make it clearer.

**Response**: We rewrote this sentence:

" *Equation (16) or (17) can be used to analyze the motivating assumption and the two basic experimental assumptions that underlie MMRT. First, Hobbs et al. (2013) suggested that MMRT was motivated by noting that enzyme denaturation cannot satisfactorily explain the temperature dependence of catalysis rates. They then assumed that all enzymes are effectively in their active state to do catalysis, and attributed the decline in enzyme catalysis rate above an optimum temperature to the change of heat capacity associated with the enzyme catalysis. However, thermally reversible enzyme denaturation, as one type of enzyme denaturation, has been observed by many studies (Sizer, 1943;Alexandrov, 1964;Huang and Cabib, 1973;Maier et al., 1955;Weis, 1981), as well as by molecular dynamics simulations (McCully et al., 2008), and is ensured to occur by the thermal motion of molecules and ions in the enzyme solution.*"

**Comment 8**: Figure 2. Given the scale used it is difficult to closely examine the fits. But the majority of the fits appear excellent (except Barnase?). I was expecting comparisons to MMRT fitted lines since that is the point of the paper. For that matter, I speculate that the Ratkowsky 2005, Corkrey et al (2012) models and others would do as well and could be fitted. The fits are summarized as r2 values. The authors comment (L155-160) that uncertainties were not available, and that they were hindered by an 'ill-conditioned Hessian matrix', but I don't see why resampling could not be used to obtain 99% CIs bands (although they would be very narrow).

**Response**: We note that there were not many data points to extract from the original papers. Particularly, most of the data in those papers were plotted in log space, so we were not able to extract data uncertainty meaningfully. We thought about resampling, but with the few data points, it is problematic. Nonetheless, based on the excellent fitting we obtained, and the fact that Ratkowsky et al (2005) model is a special case of the chemical kinetics theory here, we think not presenting the uncertainty should not affect the conclusion of our analysis. In the past, we have also used the Ratkowsky model to fit similar data, and it worked well just as the reviewer pointed out.

**Comment 9**: L210-220. I found this enlightening. Please elaborate on what was flawed in La Mer (1933).

**Response**: We explain this below. Specifically, in that paper, La Mer suggested that the rate constant should be $k_f = Z_f \cdot exp(-\Delta G_f/RT)$ for forward reaction, and $k_b = Z_b \cdot exp(-\Delta G_b/RT)$ for backward reaction, so that the equilibrium constant is $K_e = k_f/k_b =$

$Z_f/Z_b \cdot exp(-\Delta G^o/RT)$, with $\Delta G^o = \Delta G_f - \Delta G_b$ (being a linear function of temperature). Because (La Mer, 1933) assumed a non-zero heat capacity for the free energy of activation, one will find $Z_f \neq Z_b$, whereas $Z_f = Z_b$ is needed to make $K_e = exp(-\Delta G^o/RT)$, so that it is consistent with the transition state theory and thermodynamics. However, per suggestion from the other reviewer, we removed this paragraph from the revised text.

**Comment 10**: L230-235. The point made about Ohm's law is well made. This an important point since several of the references made (e.g. Corkrey et al 2012) refer to organismal growth rates and not enzymic data. While it might be argued that the complexity of processes involved at the organism precludes such models being successful, this has been found to be incorrect, such in the above reference. Note that Corkrey et al (2014; 10.1371/journal.pone.0096100) may be relevant here since it also refers to multicellular strains. It is such extensions from the enzymic to large scale processes that makes the paper relevant to the journal.

**Response**: We are glad this point resonated with the reviewer's thoughts. We now related to more references, including Corkrey et al. (2012, 2014). We also highlighted that the success of Ghosh et al. (2016) using proteomic data to explain the temperature dependent growth rate of mesophilic and thermophilic bacteria suggests that the chemical kinetics theory is scalable.

---

## Author Response (AR2)

**Response letter**

**Comment from the Handling Editor**

   One the two former referees has reviewed the revised version of your manuscript. He found that despite your detailed responses you have not really modified the content of the manuscript to fit with the main message it seems to convey: the theoretical model you developed is better than the other models. I must agree that this statement lacks of empirical (statistical) demonstration.

   I propose you to re-submit a revised version of your manuscript where you present your theory as an alternative approach to model the temperature response of enzyme activities. You can say that this theory can solve some limitations of the MMRT approaches but further works are required to verify whether this alternative approach provide reproduce observed patterns than the MMRT approaches. After submission, this revised version will be sent to a new reviewer specialist in enzyme kinetics.

   Of course, you can decide to not re-orientate the message of your manuscript and submit it elsewhere. Please inform me of your decision

Regards,

Sébastien Fontaine

**Response**: Dear Prof. Fontaine, many thanks for your patience in handling our manuscript. Based on you suggestion, we have rewritten our manuscript that specifically focuses on the chemical kinetics theory, while minimizing criticisms on MMRT. Since the manuscript has changed so significantly, we did not submit a track-change version. For the same reason, we only responded to questions from the last reviewer that are sufficient generic and potentially of interest to the new reviewer of this rewritten manuscript.

**Comment 3**. Previous comment about a bulge in the data Figure 4 authors response "However, it was also caused by the bug in the plotting script. The corrected Figure 3d is smoother." The bulge is still very evident in the resubmitted version and is derived from their equilibrium equation. Do the authors believe this is reasonable fit? it does not look like enzyme data to me.

**Response**: Figure 3d is a theoretical prediction based on inferred parameters from Figure 2k, where the model fitting is found very smooth and accurate.

**Comment 5**. No errors are given for the parameters derived from the fitting of the equilibrium model are given and the authors state that this is because they could not get errors of the data that they extracted. I would have though the error of parameters that they were interested in not in the variability of individual points but rather the error associated with fitting their 4-parameter equation to data. There is a reasonable number of enzyme temperature response data that can be collected from the literature or indeed by the authors to formally test their ideas. But if we accept where they have reached this does not allow the authors to make assertions about a better model from a fit perspective if they don't have errors or AIC comparisons.

**Response**: We explained in the revised manuscript that because "fminsearch" always obtained the same values for the best-fitting parameters even by starting from different initial guesses, it made it difficult to quantify the uncertainty by running the model fitting multiple times. For other reasons explained in section 2.3, we were not able to obtain meaningful uncertainty by Monte-

Carlo method, the finite difference method, or the bootstrapping method. In addition, we note that the Ratkowsky model, a special case of the chemical kinetic theory, has been proven very successful at fitting hundreds of published datasets (e.g., Ratkowsky et al., 2005; Corkrey et al., 2012; Ghosh et al., 2016), which testifies the merit of our theory here. More importantly, in the revised manuscript, we stressed that the major merit of the chemical kinetics theory is that it provides us with more mechanistic insights by its consideration of one well-known observation that thermally revisable enzyme denaturation is ensured by the ceaseless thermal motions of molecules and ions in the enzyme solution, and three well-established chemical reaction theories: (1) the law of mass action; (2) diffusion-limited reaction theory; and (3) transition state theory.

**Comment 6**. The assertion that the equilibrium model can explain why when adding substrate can lead to a lower Topt (ln 285) and an explanation for the lower Topt reported in e.g., Numa et al when glucose is added needs a fair comparison. The reported lower Topt predicted using the equilibrium model and lowering the activation energy in the resubmitted paper was from a Topt of 57° to 55°C (that is 2°C shift for a large change inactivation energy). This is not reasonable to compare this to Numa et al where the Topt of respiration from glucose (and other sample substrates including a mixture of yeast extract) amendment samples was about 35°C and without glucose was greater than the highest incubation temperature of ~52°C a downward shift of 17°C. **Response**: In the revised manuscript, we made it clear that the actual interpretation of the shift of optimal temperature requires a much more comprehensive modeling framework that is beyond this study. However, for the sing-substrate-single-enzyme reaction, we find that the substrate availability can shift the optimal temperature as much as 35 K (i.e., Figure 3c). Therefore, it is reasonable to hypothesize that our approach when included into a comprehensive model may be able to explain the optimal temperature change as observed in those experiments. We will evaluate this hypothesis elsewhere.

---

## Author Response (AR3)

We thank the reviewer for providing us with helpful comments, which are addressed below (in blue font). Line numbers refer to the revised manuscript, in which our changes are in Track Changes.

**Response to the reviewer**

**Comment**: This work extends the theory published in 2005 by Rathowsky et al to take into account the effect of substrate availability on the temperature response of enzyme activities. The theory proposed here, called the chemical kinetics theory, combines the law of mass action, von Smoluchowski's diffusion limited reaction and Eyring transition state theory. The proposed theory is confronted against data from 12 series of enzymatic assays extracted from the literature. The main result is to show that the temperature-dependent affinity parameter of enzymes to their substrate controls the response of enzymatic reaction rates to temperature. In particular, reaction temperature optimums shift towards higher temperatures as substrate availability increases. This significant scientific advance deserves to be published in Biogeosciences.

However, this work has important limitations that would require modifications to the article before publication.

**Response:** We appreciate your positive appraisal of our study. When revising the manuscript, we have carefully followed your suggestions, and believe that they have helped improve the manuscript significantly.

**Comment**: First of all, the article risks missing its objective if it doesn't make a greater effort of pedagogy. You are addressing a fairly generalist readership interested in many processes from different disciplines (biogeochemistry, ecology, agronomy, soil science...), not just chemists specializing in chemical kinetics. You therefore need to make a greater effort to introduce the various concepts and equations that could perhaps be basic for chemists. For example, the differences between Gibbs energies, enthalpies and the link with the heat capacity of protein unfolding/refolding, and the linear and non-linear responses that follow. The introduction of several equations is done simply by quoting another study without any real explanation. This makes it very difficult to follow the paper without reading twenty or so articles in parallel.

**Response:** We now write out and explain the equations of those basic relationships explicitly to improve the readability of our revised manuscript. For example, (1) we revised the caption of Figure 1 to reflect the relationship between Gibbs free energy, enthalpy and entropy involved in the transition state theory applied to the forward conversion of enzyme-substrate complex into products. The same information is also highlighted at Lines 130-131. (2) We revised equation (10) to show the relationship between the heat capacity of enzyme unfolding with the enthalpy and entropy involved in enzyme unfolding, and explained in Line 142 that the heat capacity of enzyme unfolding is computed as the partial derive of enthalpy with respect to temperature.

**Comment**: Furthermore, in my opinion, it is necessary to create a table summarizing all the variables and parameters, including definitions and units. The Ent variable is introduced in the equation, but this variable, which seems important, no longer appears in subsequent equations. I was wondering whether the presentation of gross equations before their versions with a standard (reference) temperature would be necessary to better understand the demonstration.

**Response**: We have created a nomenclature table in the revised appendix. We also carefully checked the presentation of equations, and made updates that help improve clarity and readability.

**Comment**: It would also be very useful to construct a table summarizing the parameters that have been adjusted on the basis of experimental data, and their values.
**Response:** We have reported the inferred parameters ($T_H$, $T_S$, $\Delta H_V$, and $\Delta C_p$) for each dataset in Figure 2.

**Comment**: In the end, how many parameters are needed to model these results? I'm amazed at the impossibility of obtaining parameter uncertainties, despite what appears to be a substantial data set.
**Response:** The model requires inferring four parameters for each enzyme essay data (as shown in each panel of Figure 2) and this information is noted at Line 150. As we discuss in section 2.3 (Lines 159-181), without uncertainty information from the original data, and due to the ill-condition of the Hessian matrix of the inference problem, we are not able to make a meaningful uncertainty estimation.

**Comment**: The concept of quasi-steady-state-approximation for the equation is unclear. What have you done mathematically? What are the "biological" assumptions behind this choice?
**Response**: We added an explanation to the revised manuscript (Line 95) along with an equation that illustrates the concept: i.e., $k_1^+ E_n S = (v_{max} + k_1^-)C$, with $C$ being the concentration of enzyme-substrate complex $E_n S$. Basically, this assumption means the concentration of enzyme-substrate complex is in rapid equilibrium during its formation and destruction. This assumption places some constraint on the kinetic parameters. Quasi-steady-state-approximation (QSSA) is the standard assumption in deriving Michaelis-Menten kinetics. QSSA was used by (Michaelis and Menten, 1913), and its rich content was discussed thoroughly in (Borghans et al., 1996).

**Comment**: I think a limitation of this work is to consider only temporary and reversible inactive forms of enzymes. However, the incessant movement of molecules inexorably leads to the definitive denaturation of enzymes. This denaturation is very often rapid (within a few hours) with important consequences for the temperate effect on enzymatic activities and living organisms (see Alvarez et al. 2018). This limit should be discussed.
**Response**: We now highlight the importance of irreversible enzyme denaturation by citing (Alvarez et al., 2018) in the introduction (Line 77) and discussion sections (Lines 226-227). We also note that (1) a dynamic model should consider both production and destruction of enzymes, and (2) the ReSOM model (Tang and Riley, 2015) applies the chemical kinetics theory, and considers irreversible enzyme denaturation.

**Reference**
Alvarez, G., Shahzad, T., Andanson, L., Bahn, M., Wallenstein, M. D., and Fontaine, S.: Catalytic power of enzymes decreases with temperature: New insights for understanding soil C cycling and microbial ecology under warming, Global Change Biol, 24, 4238-4250, 10.1111/gcb.14281, 2018.

Borghans, J. A. M., DeBoer, R. J., and Segel, L. A.: Extending the quasi-steady state approximation by changing variables, B Math Biol, 58, 43-63, Doi 10.1016/0092-8240(95)00306-1, 1996.

Michaelis, L., and Menten, M. L.: The kinetics of the inversion effect, Biochem. Z., 49, 333-369, 1913.

Tang, J. Y., and Riley, W. J.: Weaker soil carbon-climate feedbacks resulting from microbial and abiotic interactions, Nat Clim Change, 5, 56-60, 10.1038/Nclimate2438, 2015.